# Plastics in Heritage Science: Analytical Pyrolysis Techniques Applied to Objects of Design

**DOI:** 10.3390/molecules25071705

**Published:** 2020-04-08

**Authors:** Jacopo La Nasa, Greta Biale, Barbara Ferriani, Rafaela Trevisan, Maria Perla Colombini, Francesca Modugno

**Affiliations:** 1Department of Chemistry and Industrial Chemistry, University of Pisa, 56124 Pisa, Italy; jacopo.lanasa@for.unipi.it (J.L.N.); greta.biale@gmail.com (G.B.); maria.perla.colombini@unipi.it (M.P.C.); 2Barbara Ferriani S.r.l., 20122 Milano, Italy; barbaraferriani1@gmail.com; 3Conservation Laboratory at Triennale Milano, Museo del Design Italiano, 20121 Milano, Italy; rafaela.trevisan@triennale.org

**Keywords:** plastic, heritage, design objects, pyrolysis coupled with gas chromatography and mass spectrometry (Py-GC/MS), evolved gas analysis coupled with mass spectrometry (EGA-MS)

## Abstract

The first synthetic polymers were introduced as constituents of everyday life, design objects, and artworks at the end of the 19th century. Since then, the history of design has been strictly connected with the 20th century evolution of plastic materials. Objects of design from the 20th century are today a precious part of the cultural heritage. They raise specific conservation issues due to the degradation processes affecting synthetic polymer-based plastics. Museums and collections dealing with the conservation of design objects and modern materials need to base their conservation strategies on compositional data that reveal the formulations of historical plastics and their decay processes. Specific and specifically optimized analytical tools are thus needed. We employed flash analytical pyrolysis coupled with gas chromatography and mass spectrometry (Py-GC/MS) and evolved gas analysis coupled with mass spectrometry (EGA-MS) to characterize “historic polymeric materials” (HIPOMS) and heritage plastics at the molecular level with high chemical detail. This approach complements non-invasive spectroscopic diagnosis whenever it fails to obtain significant or complete information on the nature and the state of preservation of the materials under study. We determined the composition of several 20th century design objects (1954–1994) from the Triennale Design Museum of Milan (Triennale Milano - Museo del Design Italiano), which for different morphological, chemical, or physical reasons were unsuitable for characterization by non-invasive spectroscopy. EGA-MS proved capable for the study of the different fractions constituting heterogeneous micro-samples and for gaining an insight into their degradation processes from the contextual interpretation of thermal and mass-spectrometric data.

## 1. Introduction

Investigating modern materials in art is not limited to paintings: a wide range of polymer-based plastics are included in sculptures, and also in design objects that are often housed in important museums, collections and exhibitions [1,2]. Since the beginning of the 20th century, a huge variety of synthetic polymer formulations have been used to produce objects of everyday use and design objects, replacing the role of natural materials such as wood, metal or ivory. Today, aged synthetic or semi-synthetic plastics are often encountered in museums and collections and are often referred to as “*historic polymeric materials*” (HIPOMS, comprising natural polymeric materials industrially transformed and historic synthetic polymers in heritage objects [3]), “modern materials” or “heritage plastic”. The long-term behavior of these materials was poorly known when they were first introduced, and artists and designers were unaware that some of the first plastics, such as cellulose nitrate and acetate, poly(vinyl chloride) and the early polyurethanes, would quickly degrade [2,4,5].

Museums and collections dealing with the conservation of design objects and modern materials thus need to base preservation strategies on compositional data. However, conservators and curators do not, in many cases, have clear information on the composition of plastic objects in collections.

Specific analytical tools are thus needed to reveal the formulation of historical plastics and their decay processes. The investigation of plastics in museums is usually based on non-invasive characterization by vibrational spectroscopy, which can be performed in-situ using portable instrumentation. This approach has proven successful for many case studies [6,7,8,9,10,11,12,13,14,15,16,17,18,19,20]; however, in some cases, reflectance vibrational spectroscopy only supplies partial chemical information on the composition of an artwork or of a design object [18,19,20]. This is particularly true when:The morphology, the surface texture or the color prevent surface spectroscopic analysis;The surface composition is not representative of the bulk;Spectroscopy is not selective enough to resolve mixtures of different organic materials;The identification of the specific formulations is targeted (plasticizers, organic pigments, specific monomers and co-monomers, etc.);The information on degradation processes such as cross-linking or depolymerization is needed.

Laboratory micro-destructive techniques based on pyrolysis and mass spectrometry are recognized powerful techniques for analyzing polymers, polymeric mixtures and plastic formulations containing organic pigments and additives. When sampling is feasible without damaging the object under study, such techniques can play a key role in obtaining molecular information.

These approaches go beyond identifying functional groups on the basis of vibrational bands and are also applicable in cases where the morphology or surface features prevent non-invasive analysis. Analytical pyrolysis coupled with gas chromatography separation and the mass spectrometry identification of the pyrolysis products (Py-GC/MS) is one of the most powerful approaches for obtaining a complete picture of the composition of a polymeric material using a minimum amount of sample (10–100 μg) [21,22,23,24,25,26,27,28,29].

In this paper, we present the potential of analytical pyrolysis in the analysis of plastic design objects through selected case studies. These case studies highlight the advantages of using analytical pyrolysis when non-invasive spectroscopic analysis did not allow the obtaining of significant or complete information on the nature and the state of preservation of the materials under study.

Py-GC/MS maximizes the information obtained from the analysis of a micro-sample. It characterizes not only the occurrence of specific polymers, but also the presence of different additives, such as plasticizers and synthetic organic pigments, thanks to the gas chromatographic separation of the pyrolysis products, that are identified on the basis of their mass spectra. The degradation of plastic materials is, at present, a crucial conservation issue in heritage science [30,31,32], and pyrolysis-based analytical techniques have a great potential to contribute to gaining information on the state of preservation of polymers [27]. Another instrumental asset for analytical pyrolysis is Evolved Gas Analysis-Mass Spectrometry (EGA-MS), which involves the thermal separation of the materials in a sample, by slowly heating the sample in the absence of oxygen. The analytes derived from the desorption/pyrolysis processes are directly analyzed by the mass spectrometric analyzer without chromatographic separation. This approach achieves information complementary to that from the Py-GC/MS analysis, such as the thermal features of the material, the decomposition temperature of the various fractions (as in a thermal analysis) and the extent of reticulation and depolymerization [33,34,35].

Applying both analytical approaches on an amount of material weighing less than 100 µg enables analysts to perform both approaches for one single micro-sample. This thus maximizes the information obtained—with minimum invasiveness—and provides a complete picture of the composition of the sample.

We explored this approach for the analysis of eight design objects (1954–1994) from the collection of the Triennale Milano (Milan), the most important museum in Italy dedicated to design. Part of the work was performed within the framework of the project “Plastic Materials in the Collection of the Triennale Design Museum” (2017) founded by the IPERION CH.it research infrastructure. The study represents the first extensive application of analytical pyrolysis to the characterization of musealized design objects.

The objects examined were selected either because they showed conservation problems that were intrinsic features of the aging of the material, or because the artefacts had suffered mechanical damage. The characterization of the constituent materials was necessary in order to design targeted interventions or to define appropriate conservation models, as well as to obtain basic information on the museum items. Such knowledge of the museum collection not only helps the restorer-conservators but also provides a true and inexhaustible heritage for scholars, designers and historians. In order to understand the heterogeneity of the selection of objects, it is also important to underline that the results presented here are part of a wider plan of diagnostic campaigns aimed at covering many items in the Triennale’s design collection.

Each object (described in Section 2.1 and Table 1) had particular preservation issues: the rubber of the fan VE505 (1953), for example, shows a loss of elasticity, exudation and in some areas, embrittlement and decohesion. The polyurethane of Contenitoreumano is part of a very complex multi-material artefact of which very few copies have been made, all of which are now missing except for this polyurethane foam [33] that is the only ‘original’ element left from 1969.

IN 301 by Angelo Mangiarotti is a seat made of an unusually ‘rigid’ foam. Again, very few copies were made, and it is affected by major structural damage for which it is very difficult to define a plan of action.

Finally, we also focused on the techniques and methodologies used in the work of Gaetano Pesce, whose production is characterized by artisanal and artistic experimentation with synthetic resins.

None of the objects could be successfully or completely characterized by non-invasive spectroscopic techniques for different reasons, which are explained for each object in Section 3. Micro-samples were collected from the objects with negligible damage. Thus, the case studies presented here are discussed as a methodological validation of an investigation of plastic objects that explores the complementarities of spectroscopic and thermal analyses.

## 2. Results and Discussion

This section reports the main results obtained by EGA-MS and Py-GC/MS, analyzing micro-samples collected from the objects of study, discussed in relation to their morphological and chemical-physical proprieties.

### 2.1. Surface Morphology

The morphology of the surface of an object plays a crucial role in performing a non-invasive spectroscopic analysis. The “Angel Lamp” by Gaetano Pesce (1994) is a good example of an object with a morphology that prevents reflectance spectroscopy analysis from being performed. The lamp is made up of tightly entangled plastic filaments. In the top spherical part that forms the “head” of the “angel,” the filaments were melted together. Due to the structural conformation of the filaments, which prevented a plane surface from being examined (Figure 1), it was not possible to directly perform in-situ infrared spectroscopy analysis.

The Py-GC/MS analyses were performed on three different samples taken from the object: one taken from an area far from the lamp bulb, the second one taken next to the bulb holder, and the last one taken from the head of the lamp. The pyrolytic profiles of the three samples are reported in Figure 1.

The pyrograms of the three samples were all characterized by the presence of 2-ethyl hexane, 2-ethyl-1-hexanol, phthalic anhydride and hexyl benzoate. The presence of these species, together with short chain alkenes, suggested that the plastic filaments used for the production of the lamp were made of polycyclohexylenedimethylene terephthalate (PCT), a thermoplastic polyester produced until 2010 [36]. The Py-GC/MS analysis also enabled us to highlight the presence of several plasticizers, and to specifically identify diisooctyl phthalate and diisodecyl phthalate as the most abundant.

The use of pyrolysis also enabled us to obtain detailed information on the degradation processes occurring in the polymers. In this specific case study, a comparison of the chromatograms obtained for the three samples highlighted the different degrees of loss of plasticizer in different areas of the lamp. In detail, plasticizer additives were preserved in the sample area far from the light bulbs; they presented a lower abundance in the sample next to the bulbs and were completely absent in the head of the lamp. These results suggest that the loss of plasticizer was directly related to the proximity to the light sources and to the nature of the plasticizers. In fact, the heat locally emitted from the lamp bulbs promotes the evaporation of these additives, which are characterized by relatively high volatilities. The loss of plasticizer caused alterations in the physical properties of the PCT resulting in fragility of the material.

In order to obtain complementary information on the preservation state of the object, EGA-MS analyses were performed on two samples: a sample from the side of the lamp, far from the light source; and one from the head of the lamp, which consisted of melted plastic. Figure 2 reports the overlaid EGA-MS curves of the two samples: the one from the side of the lamp in black, and the one from the head of the lamp in blue.

For the sample from the side of the lamp, three thermal decomposition zones were detected: the first from 180 °C to 290 °C, with a peak at 240 °C; the second from 290 °C to 350 °C; and the third from 350 °C to 550 °C, with a peak at 390 °C. On the other hand, only two thermal degradation zones were detected for the sample from the head of the lamp: one from 180 °C to 350 °C, with a peak at 250 °C; and the other from 350 °C to 550 °C, with a maximum at 370 °C, which corresponds, on the basis of the mass spectra, to the third thermal degradation zone of the other sample. The mass spectra indicate that the first zone is related, for both samples, to the desorption of additives and to the first thermal degradation step of the polymer, since the most abundant ions in the mass spectra of this region were fragments with *m*/*z* 279, 167, 149, 57 and 43, characteristic of the mass spectra for diisooctyl phthalate. The second thermal decomposition step for the sample from the side of the lamp may also be related to the desorption of additives and to the partial depolymerization of the polymeric network, since the most abundant ions were fragments with *m*/*z* 307, 167, 149, 57 and 43. The most abundant ions in the last thermal degradation step were *m*/*z* 149, 117, 101, 87, 59 and 43, produced in the complete pyrolysis of the polymeric network, while the ions 307 and 167 can be attributed to the desorption of diisodecyl phthalate.

Comparing the two EGA profiles, the EGA curve of the sample from the head of the lamp showed only two degradation steps and also showed a 20 °C decrease in the degradation temperature of the polymeric network. This may be linked to the degradation promoted by the proximity to the light source in the sample from the head of the lamp, which caused the network to depolymerize. Since fragments with *m*/*z* 149 are common to both the pyrolysis products of phthalates and those of the terephthalate polymer [31], to obtain a more detailed description of the thermal degradation behavior of the different components, significant characteristic ions were extracted from the mass spectral data of the EGA profiles.

Figure 3a,b report the fragmentograms of the samples from the side and head of the lamp, respectively. The ions extracted were *m*/*z* 307, which corresponds to phthalates and, in particular, to diisodecyl phthalate (one of the two most abundant plasticizers identified by Py-GC/MS); and *m*/*z* 59, which corresponds to the main pyrolysis product of the polymeric network. The main difference between the two thermograms is the reduction in the abundance of phthalates (*m*/*z* 307), which is consistent with the findings highlighted by Py-GC/MS.

### 2.2. Sticky Surfaces

Another issue to be considered is the stickiness of the surface: materials with good tacky properties, such as adhesives, are generally difficult to analyze with contact spectroscopic analysis, such as attenuated total reflectance (ATR) infrared spectroscopy. A good example of the analysis of this class of materials in heritage science is of the adhesive in “Contenitoreumano n.1” by Ico Parisi and Francesco Somaini (1968). This object is made up of polyurethane (PU) foam parts joined together with an adhesive. The polyurethane foams were previously studied by analytical pyrolysis and evolved gas analysis [33], proving the potential of these approaches in the study of this class of materials. The results on the composition of the adhesives are presented here. Since the adhesive residues showed two different colorations, samples were taken from the two differently colored areas.

The Py-GC/MS chromatogram obtained for the first adhesive sample is shown in Figure 4a. The pyrolytic pattern was characterized by the presence of chloroprene, 1-chloro-3-methylbenzene, and 1-chloro-4-(1-chlorovinyl)-cyclohexene, together with benzene derivatives and high molecular weight species, characterized by a mass spectrum with *m*/*z* 239 as the base peak ion. The presence of these pyrolysis products may be related to polychloroprene, a polymer that was first produced in 1932 and is used in the formulation of contact adhesives [31,37,38].

The Py-GC/MS chromatogram obtained for the second adhesive sample is reported in Figure 4b. The pyrolytic profile is very similar to the first one and is characterized by the presence of polychloroprene. Interestingly, it also shows the presence of phenol and of high molecular weight phenolic species. These compounds may be related to the presence of a phenolic resin in the formulation, another synthetic organic material commonly used in the production of adhesives. [31]. The profile of the phenolic pyrolysis products was also carefully compared with the pyrolysis of reference organic pigments known to produce phenolic pyrolysis products such as β-naphthol, in order to rule out the possible derivation of such pyrolysis products from pigments [39,40].

These results showed the presence of two different adhesive formulations in the same objects, suggesting the use of two different commercial materials in the production.

### 2.3. Surface Color

Light absorption from dark surfaces prevents reflectance spectra with a good resolution from being obtained [17], as in the black polymeric parts of the fan “VE 505” by Ezio Pirali (1953) shown in Figure 5.

The pyrolytic profile obtained for the sample of the front gasket is reported in Figure 5a, and shows that the main pyrolysis products were isoprene, limonene, xylene, styrene and several high molecular weight aliphatic compounds. These pyrolysis products point to the use of isoprene-styrene rubber to produce the gasket. The analysis also highlighted the presence of several plasticizers, with isobutyl phthalate, butyl phthalate and bis(2-ethylhexyl)phthalate as the most abundant, suggesting the synthetic nature of the polymer [31,38].

The Py-GC/MS chromatogram obtained for the sample from the power cable is reported in Figure 5b. The pyrolytic profile of this sample was characterized by the presence of alkene and alkane peak clusters in the range C_7_-C_26_. The results are compatible with the presence of polyethylene. The information obtained on these components is crucial to define the best conservation and restoration strategies for the object [31,38].

The use of analytical pyrolysis is also crucial for the characterization of objects that, in addition to dark surfaces, feature porous morphologies, such as polyurethane foams, which are even more complex than flat surfaces to analyze by non-invasive spectroscopic approaches. A good example of this type of material is the “IN 301” chair by Angelo Mangiarotti (1969), which is entirely produced using an unknown black foam.

The pyrolytic profile obtained for the sample of foam constituting the chair body is shown in Figure 5c. The chromatogram was characterized mainly by pentanal derivatives, alcohols and 4,4-methylene-(bis)-2-chloroaniline (MObCA), a common curing agent used in the production of polyurethanes [41,42]. The presence of MObCA, together with 2-chloroaniline and 1-chloro-4-isocyanato-benzene, suggested that the material used to produce the chair was a polyether-based polyurethane [43].

### 2.4. Surface Not Representative of the Bulk

Another limitation of surface analysis is related to the impossibility to obtain complete information on objects made up of several superimposed layers. The “Dalilauno” chair designed by Gaetano Pesce in 1980 is a good example of the drawbacks derived from limiting material investigation to the surface of an object. A complete picture of the composition of the materials used in the production of the “Dalilauno” chair was obtained by analyzing two different samples using Py-GC/MS, one taken from the glossy surface and one from the bulk of the material constituting the body of the chair. The pyrolytic profiles for the two samples are reported in Figure 6.

The pyrogram of the sample from the surface (Figure 6a) features phenol, cresol, bisphenol and high molecular weight phenolic structures: these pyrolysis products are typical markers of a phenol-formaldehyde resin [31,38].

The body of the chair was produced using a foam that could not be analyzed using a non-invasive approach due to the morphology of the materials that were highly porous. The pyrolysis profile of the sample taken from the bulk of the chair (Figure 6b) features aniline, cyclohexyldimethylamine, tripropylene glycol and various high molecular weight esters. The presence of tripropylene glycol and cyclohexyldimethylamine, used as the initiator and catalyst for urethane synthesis respectively, suggested that the body of the chair was produced using a polyester-based polyurethane foam [44].

The Py-GC/MS results suggested that the designer used polyurethane foam for the body of the chair, probably from a mold, while the phenol-formaldehyde resin was then applied as a coating to create the harder glossy surface.

### 2.5. Mixtures of Different Organic Materials and Evaluation of Degradation Processes

Another scenario where Py-GC/MS is crucial to achieve enough chemical selectivity is the analysis of mixtures of organic materials or of minor components.

The first case study is a series of chairs named “Nobody’s Perfect” by Gaetano Pesce. The Py-GC/MS chromatograms obtained for samples from the three chairs are reported in Figure 7. The pyrolytic profile of the sample from Nobody’s Perfect 1 (Figure 7a) was characterized by benzoic acid, butyl benzoate and several peaks at high retention times, which are attributable to carbonic acid di-hexadienyl ester derivatives, all pyrolysis products typical of polycarbonate [38]. The additional presence in the pyrolytic profile of 5-isocyanato-1-(isocyanatomethyl)-1,3,3-trimethyl-cyclohexane together with 4,4’-methylenedianiline (MDA) and ether oligomers suggested that the material used to produce the chair was a polycarbonate with added polyurethane [45].

The Py-GC/MS analyses performed on Nobody’s Perfect 2 from the collection showed a similar composition to the chair described above, characterized by the presence of both polycarbonate and polyurethane.

On the other hand, the pyrolytic profile of the sample from Nobody’s Perfect 3 (Figure 7b) was only characterized by the presence of the pyrolysis products derived from the polycarbonate.

The results obtained by Py-GC/MS highlighted the different material compositions of the three chairs. EGA-MS was then applied to explore the possibility of better characterizing and differentiating the components based on their thermal degradation behaviors. The EGA profiles of samples from Nobody’s Perfect 1 and Nobody’s Perfect 3 are shown in Figure 8a,b, respectively.

Four thermal degradation zones were detected for the sample from Nobody’s Perfect 1: the first one from 90 to 230 °C, corresponding to the desorption of additives, since the most abundant ions were fragments with *m*/*z* 223 and 149. In the second zone, from 230 to 330 °C, the most abundant ions were fragments with *m*/*z* 254, 225, 139 and 81, suggesting the thermal degradation of the polycarbonate fraction of the polymeric network. Meanwhile, the third zone, from 330 to 440 °C, is likely related to the thermal degradation of the polyurethane fraction, since the most abundant ions were fragments with *m*/*z* 198, 117 and 56. The last one from 440 to 550 °C, showing ions with *m*/*z* 254, 225, 106 and 59, relates to the complete pyrolysis of the polymeric network.

The EGA profile of Nobody’s Perfect 2 was very similar to the EGA profile of the first chair, considering that the constitutive material was identified as polycarbonate with added polyurethane, like for the first chair.

Three thermal degradation steps were identified for Nobody’s Perfect 3. In the first one, from 100 to 230 °C, the most abundant ions were fragments with *m*/*z* 223 and 149, which correspond to the desorption of dibutyl phthalate. The other two, from 230 to 380 °C and from 380 to 470 °C, relate to the partial degradation and the complete pyrolysis of the polycarbonate network, since the most abundant ions were fragments with *m*/*z* 254, 225 and 81 and fragments with *m*/*z* 129 and 71, respectively.

Figure 8a reports the fragmentogram of Nobody’s Perfect 1: the ions extracted were *m*/*z* 223, which refers to dibutyl phthalate; *m*/*z* 81, which corresponds to the characteristic ion fragment of carbonic acid di-hexadienyl ester derivatives; and *m*/*z* 198, which corresponds to MDA, one of the main precursors of polyurethane foams, also identified by Py-GC/MS. The fragmentogram of Nobody’s Perfect 3 is reported in Figure 8b. The ions extracted were *m*/*z* 223, *m*/*z* 81 and *m*/*z* 129, which is one of the pyrolysis products of the polymeric network identified as polycarbonate.

## 3. Materials and Methods

### 3.1. Samples

The samples from the Triennale Design Museum were a selection of the objects studied within the framework of the project “*Materiali plastici nella collezione del Triennale Design Museum*”, founded by the research infrastructure IPERION CH.it in 2017–2018, in which a series of objects selected by the museum were investigated by both non-invasive spectroscopic and micro-destructive analytical approaches [46]. The list of the investigated objects in this work is reported in Table 1.

### 3.2. Analytical Pyrolysis-Gas Chromatography/Mass Spectrometry (Py-GC/MS)

Py-GC/MS analyses were performed using a multi-shot micro-furnace pyrolyzer, EGA/PY-3030D (Frontier Lab, Fukushima, Japan), coupled with a 6890N gas chromatography system with a split/splitless injection port and combined with a 5973 mass selective single quadrupole mass spectrometer (Agilent Technologies, Santa Clara, CA, USA) [33,47,48].

The samples (80–110 µg) were analyzed by setting the pyrolysis temperature to 600 °C and the Py-GC interface to 280 °C.

The GC injector was operated in split mode with a split ratio of 1:10 and a temperature set to 280 °C. The chromatographic separation of pyrolysis products was performed on a fused silica capillary column HP-5MS (5% diphenyl, 95% dimethyl-polysiloxane, 30 m × 0.25 mm i.d., 0.25 μm film thickness, J&W Scientific (Agilent Technologies, Santa Clara, CA, USA), preceded by 2 m of deactivated fused silica pre-column, with an internal diameter of 0.32 mm (Agilent Technologies, Santa Clara, CA, USA). The chromatographic conditions were 40 °C held for 5 min, followed by a 10 °C/min ramp to 310 °C, which was held for 20 min. The helium (purity 99.9995%) gas flow was set in constant flow mode at 1.2 mL/min. The mass spectrometer was operated in EI positive mode (70 eV, scanning *m*/*z* 35–700). The mass spectrum interpretation was performed using NIST 8.0 reference library, (National Institute of Standards and Technology National Gaithersburg, MD, USA) and by comparison with the literature.

### 3.3. Evolved Gas Analysis-Mass Spectrometry (EGA-MS)

The instrumentation used for the EGA-MS analyses was the same as that used for the Py-GC/MS described in Section 2.2, used in a different instrumental configuration: the GC column was removed, and the pyrolyzer was coupled to the mass spectrometry system with a deactivated and uncoated stainless-steel transfer tube (UADTM-2.5N, 0.15 mm i.d.× 2.5 m length, Frontier Lab) kept at 300 °C, with an inlet temperature of 280 °C. The temperature program for the micro-furnace was an initial temperature of 50 °C, followed by a 10 °C/min ramp up to 800 °C. Analyses were performed under a helium flow (1 mL/min) with a 1:20 split ratio. The micro-furnace interface temperature was automatically kept 100 °C higher than the furnace temperature, up to a maximum value of 300 °C. The mass spectrometer was operated in EI positive mode (70 eV, scanning *m*/*z* 50–700). The sample sizes for the analysis were in the range 80–110 µg [36,47].

## 4. Conclusions

The results obtained within this work show how the use of analytical pyrolysis in the two analytical configurations, Py-GC/MS and EGA-MS, is an extremely powerful tool for the analysis of plastic heritage objects. Above all, it could be exploited in those cases where sampling is an acceptable option and where the non-invasive vibrational spectroscopic approach has limited effectiveness, due to specific limiting factors. Analytical pyrolysis coupled with mass spectrometry can be used to obtain molecular information on the compositions of polymers, co-polymers, additives and mixtures, and to gain information on the degradation processes affecting plastic.

Pyrolysis-GC/MS and evolved gas analysis mass spectrometry enabled us to characterize the materials used to produce different design objects that could not be analyzed by non-invasive approaches due to the proprieties of the material or the object (porosity, multiple layers, stickiness, polymer mixtures, additives, degraded molecules, etc.), which hamper the application of an in situ analytical approach.

Py-GC/MS analysis was successfully applied to characterize a wide range of materials, highlighting the presence of specific additives, such as plasticizers, or of mixtures of materials, even in complex materials.

The use of this approach also enabled us to evaluate modifications of the composition, such as the loss of plasticizers, and to correlate them to specific risk factors, such as exposure to light and heat.

Finally, EGA-MS analyses produced useful complementary information to evaluate the level of degradation, such as the reticulation/depolymerization of the polymeric network.

In conclusion, to fully characterize an art plastic object, both analytical pyrolysis and vibrational spectroscopies in situ should be used to maximize the information obtained and, at the same time, minimize the invasiveness of the sampling.

## Figures and Tables

**Figure 1 molecules-25-01705-f001:**
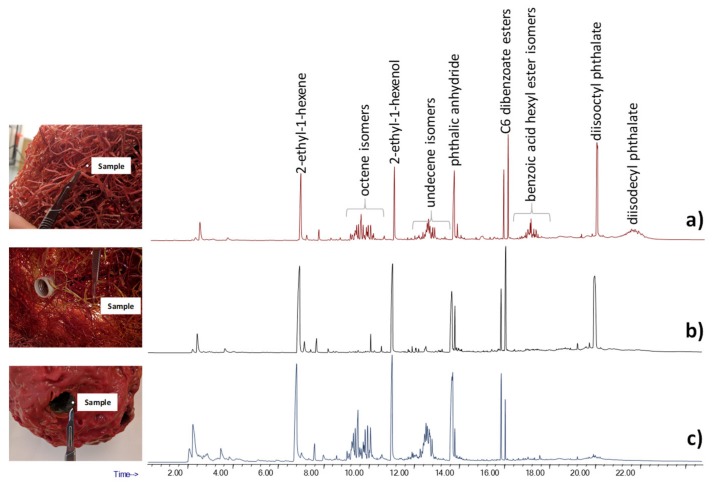
The pyrolysis coupled with gas chromatography and mass spectrometry (Py-GC/MS) chromatograms obtained for the samples from Angel Lamp: (**a**) from the side of the lamp, (**b**) next to the light bulb and (**c**) melted plastic.

**Figure 2 molecules-25-01705-f002:**
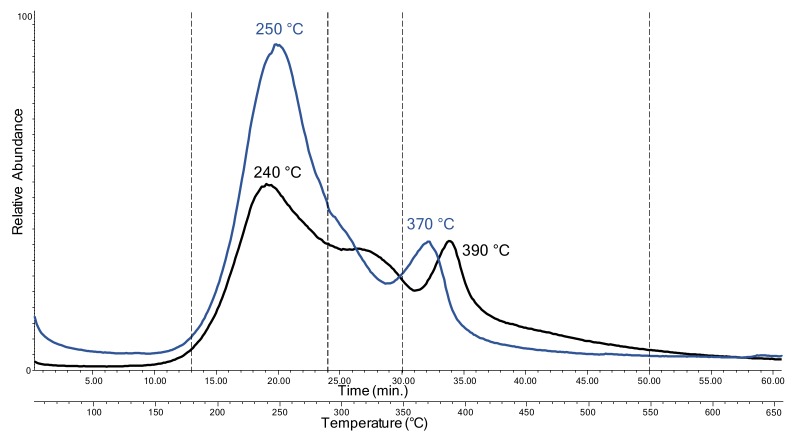
The evolved gas analysis coupled with mass spectrometry (EGA-MS) curves obtained for the samples from “Angel Lamp”: the sample from the side of the lamp (black), and the one from the head of the lamp (blue).

**Figure 3 molecules-25-01705-f003:**
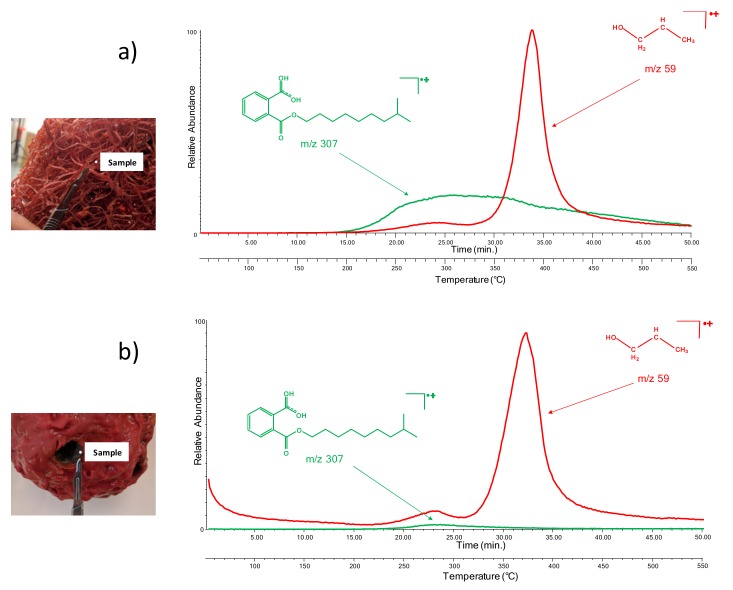
The extracted ion (*m*/*z* 307, 59) thermograms obtained for the sample from “Angel Lamp”: (**a**) the sample from the side of the lamp and (**b**) the sample from the head of the lamp.

**Figure 4 molecules-25-01705-f004:**
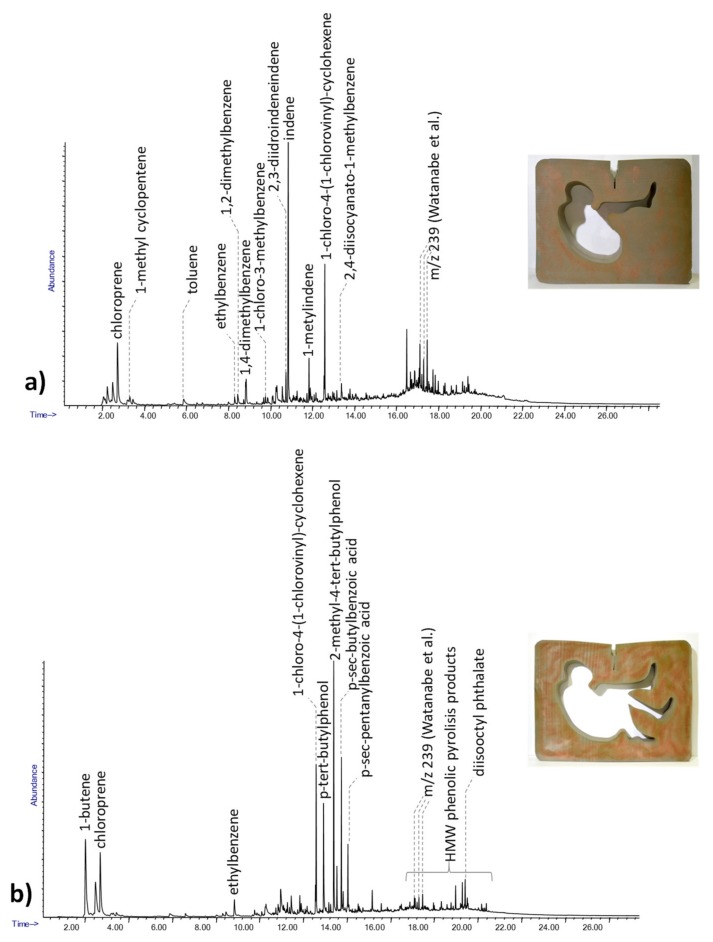
The Py-GC/MS chromatograms obtained for the white (**a**) and red (**b**) adhesive samples from Contenitore Umano. The pyrolysis products described in [31].

**Figure 5 molecules-25-01705-f005:**
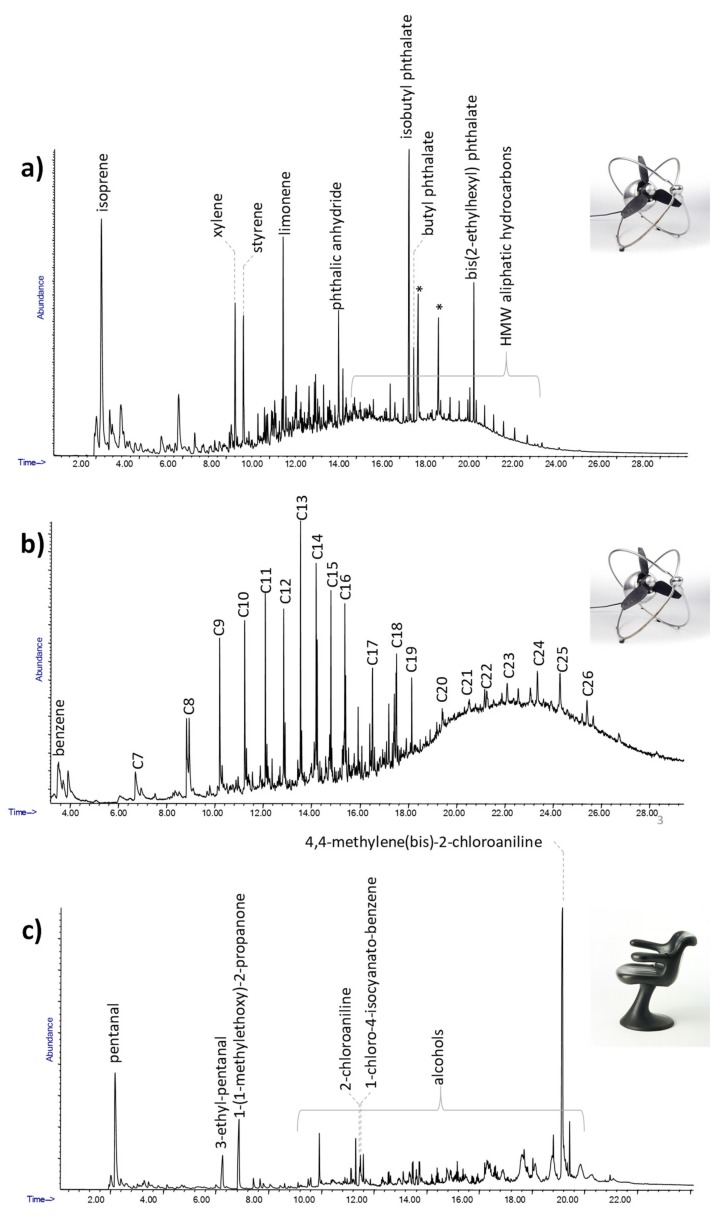
The Py-GC/MS chromatograms obtained for the samples from VE505, (**a**) and (**b**), and IN301 (**c**).

**Figure 6 molecules-25-01705-f006:**
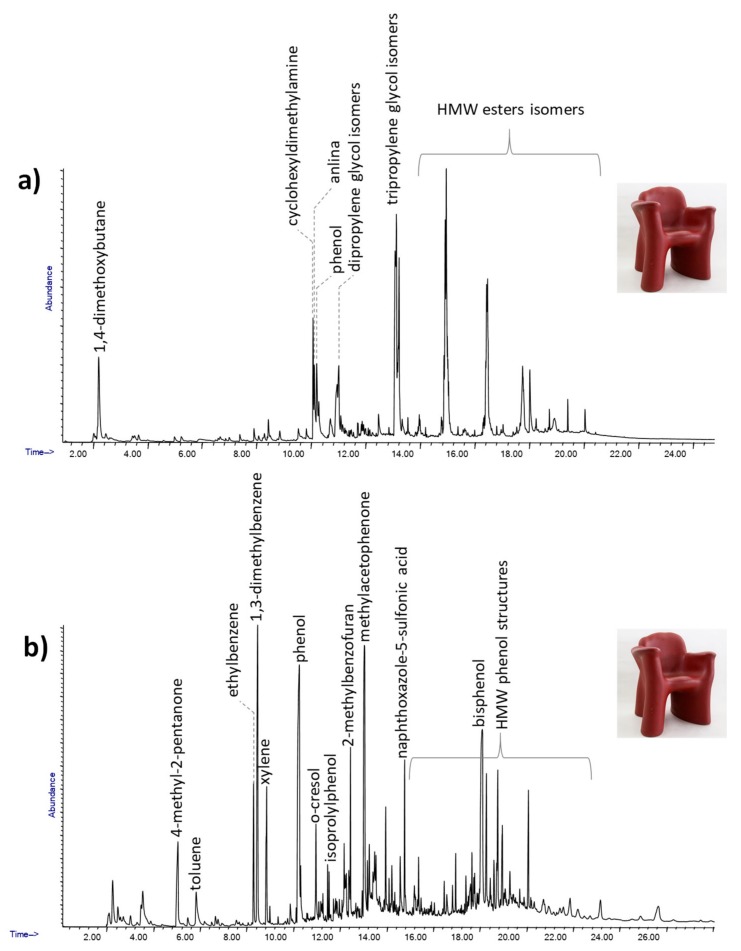
The Py-GC/MS chromatograms obtained for the surface (**a**) and body (**b**); samples from Dalilauno.

**Figure 7 molecules-25-01705-f007:**
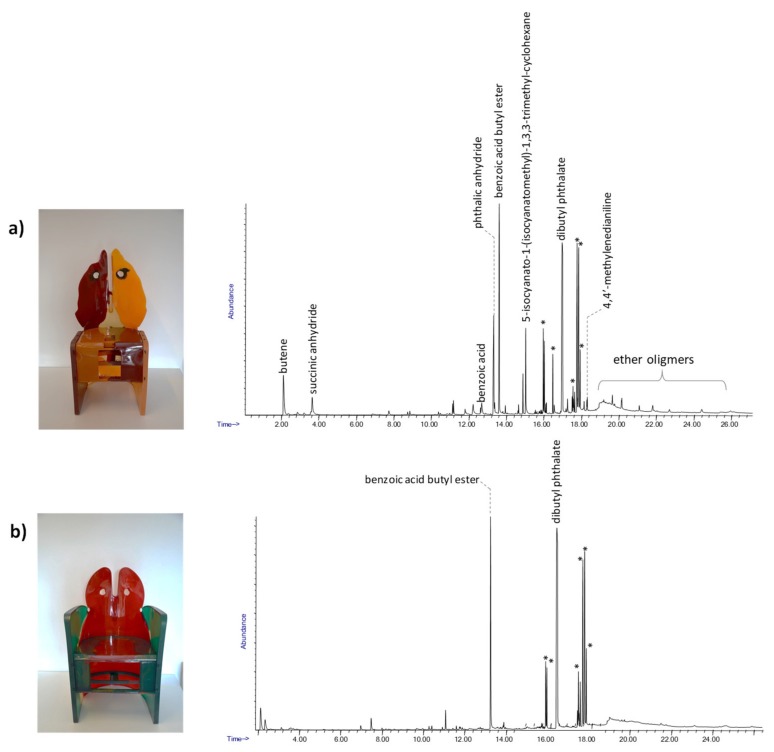
The Py-GC/MS chromatograms obtained for the chair samples: (**a**) Nobody’s Perfect 1 and (**b**) Nobody’s Perfect 3; *: carbonic acid di-hexadienyl ester derivatives.

**Figure 8 molecules-25-01705-f008:**
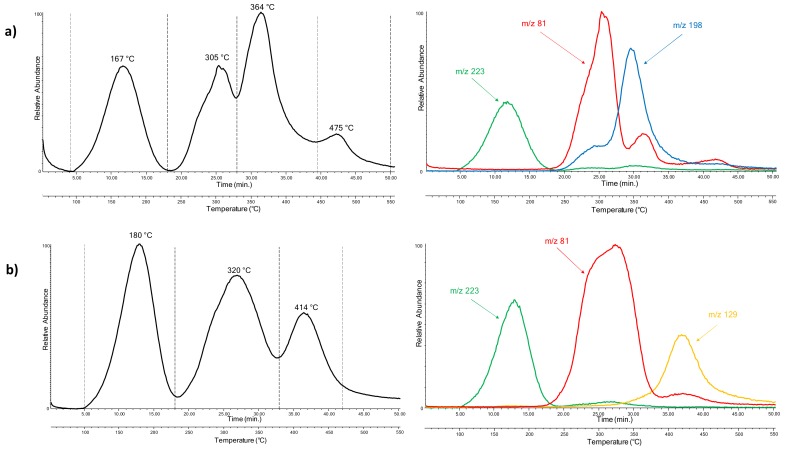
The EGA-MS curves and extracted ion thermograms obtained for samples from Gaetano Pesce chairs: (**a**) Nobody’s Perfect 1 (*m*/*z* 223, 198, 81) and (**b**) Nobody’s perfect 3 (*m*/*z* 223, 129, 81).

**Table 1 molecules-25-01705-t001:** A list of the design objects/artworks investigated, analytical issues preventing non-invasive characterization and conservation problems.

Object and Designers/Artists		Year	Issues Presented by the Object	Features that Hindered Non-Invasive Spectroscopic Analysis
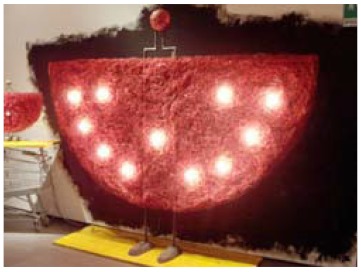	*Angel Lamp* Gaetano Pesce Manufactured by the artist (Italy)	1994	Unknown plastic	Sample morphology (the material is made of intricate 1 mm large filaments of plastic)
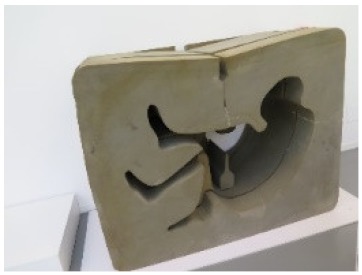	*CONTENITOREUMANO N. 1* Ico Parisi Francesco Somaini Manufactured by the artist (Italy)	1968–1969	Detachment of the glue uniting the different PU layers	Black and porous surface
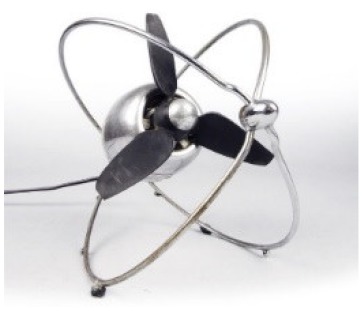	*VE 505* Ezio Pirali Zerowatt - F.E.R. (Italy)	1953	Loss of elasticity, exudation, embrittlement and decohesion	Black surface
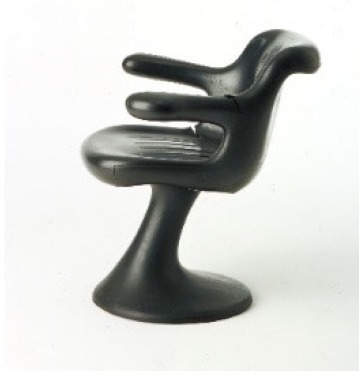	*IN 301* Angelo Mangiarotti Zanotta (Italy)	1969	Structural damages	Black and porous surface
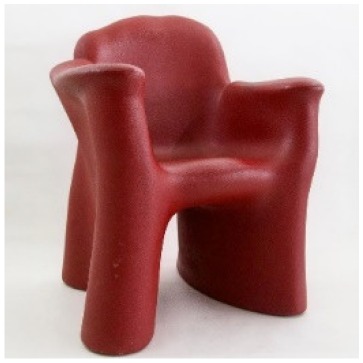	*Dalilauno* *(modello 351)* Gaetano Pesce Cassina (Italy)	1980	Unknown polyurethane type	Surface not representative of the bulk of the material
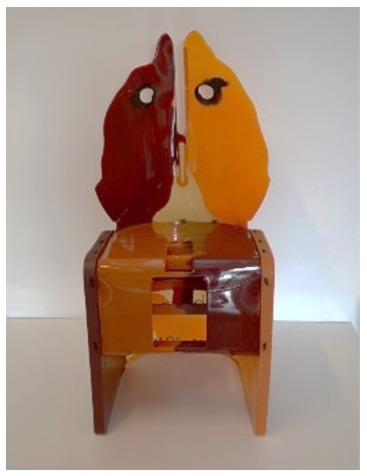	*Nobody’s Perfect 1* Gaetano Pesce Zerodisegno–Quattrocchio S.r.l. (Italy)	1993	Deformation and loss of flexibility	FTIR profile not conclusive for the identification of the material
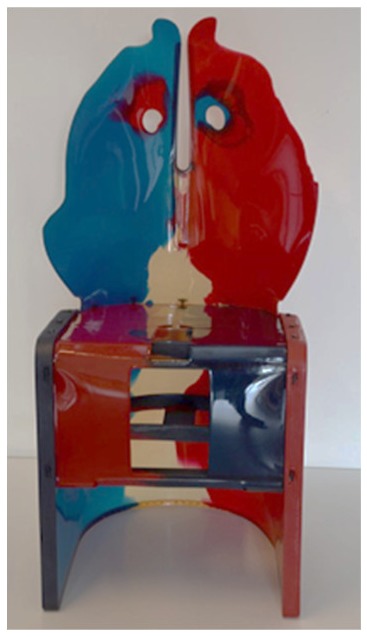	*Nobody’s Perfect 2* Gaetano Pesce Zerodisegno–Quattrocchio S.r.l. (Italy)	1993	Deformation and loss of flexibility	FTIR profile not conclusive for the identification
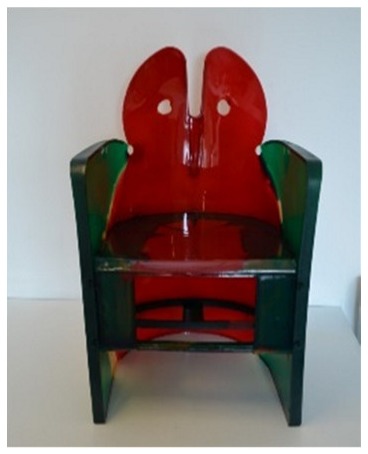	*Nobody’s Perfect 3* Gaetano Pesce Zerodisegno–Quattrocchio S.r.l. (Italy)	1993	Deformation and loss of flexibility	FTIR profile not conclusive for the identification

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
