# Peer review of "Plastics in Heritage Science: Analytical Pyrolysis Techniques Applied to Objects of Design"

_molecules, 2020, doi:10.3390/molecules25071705_

Round 1
Reviewer 1 Report
The manuscript by Modugno and co-workers is a good collection of case studies reporting the application of Py-GC/MS and EGA-MS for the identification of synthetic polymers used in design objects. Even though the description of any single case is rather general, without entering in the detail of the mechanism of pyrolysis of each single polymer (on the basis of the very large bibliography available on the pyrolysis of industrial polymers), the manuscript reads well and certainly worth the publication in this journal. Apart from some general language editing that will be carried out in the accepted manuscript, I suggest a few very minor corrections that have to be taken into account while preparing the revised version:
-p.1 line 43: polyvinylchloride should instead be poly(vinyl chloride)
-p.2 and/or 3: many objects by the Triennale's design collection were already analyzed and studied by Toniolo et al. The introduction would be incomplete without citing some of the articles published, e.g., in Polymer Degradation and Stability
-p.16: some references are incompletes or reported using an incorrect format: as examples check ref. 1 (“van” is part of the family name “van Oosten”; Oosten, T.v.” should instead be “van Oosten, T.”, ref. 16 (the paper is nor in press any longer), ref. 26, 27 and 41 (format should be correct), ref. 31 (accessed date), ref. 37 (real patent number should be add).
Author Response
We would like to thank the Reviewers for the time and effort dedicated to revising our manuscript and for the helpful advices and suggestions that we exploited to improve the paper. Detailed answer to the Reviewers' comments are included below.
Reviewer 1
The manuscript by Modugno and co-workers is a good collection of case studies reporting the application of Py-GC/MS and EGA-MS for the identification of synthetic polymers used in design objects. Even though the description of any single case is rather general, without entering in the detail of the mechanism of pyrolysis of each single polymer (on the basis of the very large bibliography available on the pyrolysis of industrial polymers), the manuscript reads well and certainly worth the publication in this journal. Apart from some general language editing that will be carried out in the accepted manuscript, I suggest a few very minor corrections that have to be taken into account while preparing the revised version:
-p.1 line 43: polyvinylchloride should instead be poly(vinyl chloride)
We amended the error.
-p.2 and/or 3: many objects by the Triennale's design collection were already analyzed and studied by Toniolo et al. The introduction would be incomplete without citing some of the articles published, e.g., in Polymer Degradation and Stability
We added references [6-10] and we improved other aspects of the bibliography, according also to the suggestions of Reviewer 2 amending some errors, eliminating a reference that was repeated twice, and substituting three references with more recent ones).
-p.16: some references are incompletes or reported using an incorrect format: as examples check ref. 1 (“van” is part of the family name “van Oosten”; Oosten, T.v.” should instead be “van Oosten, T.”, ref. 16 (the paper is nor in press any longer), ref. 26, 27 and 41 (format should be correct), ref. 31 (accessed date), ref. 37 (real patent number should be add).
We corrected the references according to the comments above and to the journal guidelines.
Reviewer 2 Report
Dear Authors,
I read with great interest your paper. The study of plastics in modern art and design objects remains a field that needs much more study, so this paper is highly appreciated. It is well structured and clearly written, although some small errors remain, a careful re-reading of the text is recommended. I only have some minor remarks.
P105: a summary of some common preservation issues are given, but only problems to some of the objects are described. I would like to suggest to include them all, or maybe better, include a column in the table 1 describing which issues there are with the objects.
P141: PCT is identified based on literature data. I was wondering if reference samples of historic plastics were analysed as well (e.g. the SamCo reference kit)? This remark is not only valid for this polymer, but for all polymers detected.
P171: ion fragments are mentioned, but it is not always clear where they are originating from, although further in the text this is mentioned to some extent. I recommend to add a table with ions and their corresponding compound identification, as this would increase the readability of the paper.
Author Response
We would like to thank the Reviewers for the time and effort dedicated to revising our manuscript and for the helpful advices and suggestions that we exploited to improve the paper. Detailed answer to the Reviewers' comments are included below.
Reviewer 2
Dear Authors,
I read with great interest your paper. The study of plastics in modern art and design objects remains a field that needs much more study, so this paper is highly appreciated. It is well structured and clearly written, although some small errors remain, a careful re-reading of the text is recommended. I only have some minor remarks.
P105: a summary of some common preservation issues are given, but only problems to some of the objects are described. I would like to suggest to include them all, or maybe better, include a column in the table 1 describing which issues there are with the objects.
Not all the objects were characterized by conservation problems, however we modified Table 1 in order to insert comments on the analysis target, and on the conservation issues presented by the objects.
P141: PCT is identified based on literature data. I was wondering if reference samples of historic plastics were analysed as well (e.g. the SamCo reference kit)? This remark is not only valid for this polymer, but for all polymers detected.
Unfortunately, we did not have access to a reference kit. Even if we did not have the access to these kits the of the pyrolysis process led to the production of specific and reproducible thermal degradation products. The interpretation of these products allows to an unambiguous identification of the polymers.
P171: ion fragments are mentioned, but it is not always clear where they are originating from, although further in the text this is mentioned to some extent. I recommend to add a table with ions and their corresponding compound identification, as this would increase the readability of the paper.
We modified the text in order to include more detailed information on the specific chemical species generating the significant ions (lines 179-188 and 344 - 355).